# Innovative approach to improve information accuracy in a two-district cross-sectional study in Bihar, India

Caroline Jeffery,[1] Marcello Pagano,[2] Baburam Devkota,[1] Joseph J Valadez  [1]

[1]Department of International Public Health, Liverpool School of Tropical Medicine, Liverpool, UK
[2]Department of Biostatistics, Harvard TH Chan School of Public Health, Boston, Massachusetts, USA

**Correspondence to**
Dr Joseph J Valadez;
joseph.valadez@lstmed.ac.uk

## ABSTRACT

**Objective** Combine Health Management Information Systems (HMIS) and probability survey data using the statistical annealing technique (AT) to produce more accurate health coverage estimates than either source of data and a measure of HMIS data error.

**Setting** This study is set in Bihar, the fifth poorest state in India, where half the population lives below the poverty line. An important source of data, used by health professionals for programme decision making, is routine health facility or HMIS data. Its quality is sometimes poor or unknown, and has no measure of its uncertainty. Using AT, we combine district-level HMIS and probability survey data (n=475) for the first time for 10 indicators assessing antenatal care, institutional delivery and neonatal care from 11 blocks of Aurangabad and 14 blocks of Gopalganj districts (N=6 253 965) in Bihar state, India.

**Participants** Both districts are rural. Bihar is 82.7% Hindu and 16.9% Islamic.

**Primary outcome measures** Survey prevalence measures for 10 indicators, corresponding prevalences using HMIS data, combined prevalences calculated with AT and SEs for each type of data.

**Results** The combined and survey estimates differ by <0.10. The combined and HMIS estimates differ by up to 84.2%, with the HMIS having 1.4–32.3 times larger error. Of 20 HMIS versus survey coverage estimate comparisons across the two districts only five differed by <0.10. Of 250 subdistrict-level comparisons of HMIS versus combined estimates, only 36.4% of the HMIS estimates are within the 95% CI of the combined estimate.

**Conclusions** Our statistical innovation increases the accuracy of information available for local health system decision making, allows evaluation of indicator accuracy and increases the accuracy of HMIS estimates. The combined estimates with a measure of error better informs health system professionals about their risks when using HMIS estimates, so they can reduce waste by making better decisions. Our results show that AT is an effective method ready for additional international assessment while also being used to provide affordable information to improve health services.

## Strengths and limitations of this study

► Household survey data were captured as a stratified random sample leading to an efficient use of information.
► Administrative data comprise 100% of the available recurrent information available in the two selected districts.
► The study population is very large covering a large geographical area, reducing the likelihood that the results are pertinent only to a small group of mothers with children; results may be generalisable.
► The process for combining probability and administrative data has been assessed using a statistically principled approach prior to use in this study.
► The study is confined to two districts of Bihar, India which indicates the need for replicating the study in additional States of India and in other country settings.

following strategies that improve health and education, reduce inequality and spur economic growth. For national strategies to be effective, they must be grounded in the local context. Good-quality data to measure the prevalence of disease conditions, or the population's coverage with health services coverage, is an indispensable resource for programme managers and health policy makers to understand their context. This point is equally true in higher-income countries as it is in low-income and middle-income countries (LMICs). In these latter settings, household surveys are generally considered resource intensive, and being carried out at 3–5 years intervals, are not sufficiently frequency for many decisions. As a result, data generated routinely through the Health Management Information Systems (HMIS) is more frequently used for decision-making and for annual reviews than household surveys. HMIS data consist of routine information reported from the health facilities to districts health offices, who submit it to Ministry of Health on a monthly, quarterly, semiannual and annual basis. A key

## INTRODUCTION

The 17 Sustainable Development Goals (SDGs) were adopted by all United Nations member states in 2015 as an urgent call for action to end poverty and deprivation by

shortcoming of HMIS data is that it is only representative of those who can access health services. This shortcoming is especially problematic in LMIC where access to healthcare services is constrained by geographic, economic and social determinants, and in settings where there are competing services in the community that do not report into the HMIS.

Demand by global donors for timely and reliable data continues to increase.[1] The 60th World Health Assembly underscored the importance of robust information to strengthen health systems and policies (resolution WHA60.27),[2] and to this end, established a Health Metrics Network (HMN) partnership to aid countries improve data quality of routine information systems to enable their use for health planning and decision-making.[3–6] This was followed by other resolutions and in 2015 by a high-level summit on Measurement and Accountability for Results leading to the formation of the Health Data Collaborative (HDC) in March 2016 and supported by global partners. HDC's mandate is more extensive than that of HMN, having an ultimate objective of aiding countries to improve the quality and availability of health data and their ability to consistently and accurately report on progress towards the health-related SDGs.

The advent of electronic health records was heralded, as studies showed their association with improved clinical care, outcomes[7] and surveillance.[8] Consequently, the advancement of a computerised District Health Information System-2 (DHIS2) was implemented in 73+ countries to integrate data sources for their more rapid analyses, dissemination and use. An evaluation in Tanzania showed it improved timeliness and completeness of reporting.[8] South Africa established a dedicated HIV/AIDS information system recommending the use of the earlier DHIS for integrating different information sources[9] and as input for designing national programme strategies.[10] A 2010–2015 study in Swaziland recommended embracing electronic medical data systems to reduce the discrepancies occurring between the existing three information systems put in place for malaria elimination.[11] Recently, a similar demand was issued by practitioners working in humanitarian settings for an epidemiology and demography service to develop robust and timely information from multiple sources to establish priorities that address the population's needs.[12]

One important use of routine information systems is to monitor a population's coverage with health service, as Global Alliance on Vaccines and Immunisations has done for vaccination rates since the 1990s. They attempt to control for known weaknesses in administrative data by carrying out data quality audits by recounting, and contrasting, the number vaccinated in health centres with the number originally reported.[13]

Although measurement improves all facets of health management, HMIS related studies mainly focus on improvements in obtaining numerators.[1 5 14 15] The future also demands accurate and precise denominators, and, more importantly, a principled method for assessing their

accuracy (eg, SE) so users understand their risk when using the data. Under normal circumstances, the error present in HMIS remains unknown.

The research question we investigate in this paper: How can we measure the error in the HMIS and improve its accuracy? We address this issue by presenting an innovation called *Statistical Annealing* (annealing technique, AT). It builds on our earlier pioneering work in 2011[16] and refined in 2018[17] that provides a coverage estimator with a 95% CI by combining data from a probability survey and HMIS data. HMIS coverage estimates have not before had a 95% CI to inform users of their accuracy. We demonstrated the magnitude of HMIS error and how it can be improved by using AT.

By the early 20h century, Bowley[18] and Neyman[19] proved the importance of sampling and the estimated magnitudes of their errors. The $SE$[20] is a commonly used measure of error, which is an estimate of deviation of the sample mean from the actual population mean. The SE is used in calculating a 95% CI, which relates to the accuracy of the estimate. If survey data are combined (annealed) with HMIS appropriately they produce a single, more accurate coverage estimator.[16] However, concurrent well-designed surveys have been a surprisingly neglected sources of information for improving HMIS.[21 22] Recent efforts have used surveys to improve HMIS estimates,[23 24] although the resulting revised estimates do not have a corresponding 95% CI.

The objective of AT is to produce a principled measure of health system performance by combining existing HMIS and survey information which also produces an accompanying measure of its accuracy. This hybrid estimate is more accurate than either data source alone. Our proof of principle study, used Child Health Day (CHD) administrative data from Benin and Madagascar provided by UNICEF country offices and household survey data collected at the same time, to verify CHD coverage and the quality of CHD administrative data.[17] This refined approach to AT resulted in the production of a 95% CI for the administrative data and for the combined result. See online supplemental information 1 for the AT formulae.

The current study assesses the transferability of AT to a new setting, HMIS data from Bihar state located in northeast India. Bihar is the fifth poorest state of India where half the population lives below the poverty line. It is one of the most densely populated states (N=110 million), and has some of the weakest maternal and child health and nutrition indicators in India.[25 26] Both districts are primarily rural with a mix of upper caste and lower caste Hindu residents, and minority Muslim communities.

Also, this study is the first use of AT with true HMIS data, which in the future will be a more typical use for AT, and result in our conclusions about its global applicability. We apply AT to 10 indicators related to antenatal care (ANC), institutional delivery, and neonatal care using data from two districts, Aurangabad (N=3 695 928) and Gopalganj (N=2 558 037) and two sources of data collected for purposes other than this study. HMIS

data from the two districts and a probability survey data collected as part of an earlier assessment of maternal and child healthcare coverage in Bihar.[27] This secondary data analysis uses datasets without personal identifiers.

## METHODS

### HMIS system in Bihar

The HMIS data in Bihar consists of 318 data elements. These are reported monthly by health subcentres (HSCs) and aggregated at the block primary health centre (BPHC). The *block* is the subdistrict administrative unit in India. Staff at the HSCs tally information from paper registers and send a paper report to BPHCs where it is entered into a computerised HMIS system, and subsequently maintained electronically. District-level indicators comprise information aggregated across the block as well as data from district and referral hospitals.

### Data

To develop AT in Bihar, we used the health facility-based HMIS and probability data from a household survey in two districts, Aurangabad and Gopalganj. From the Bihar State Health Society, we obtained two sources of HMIS data: the HMIS data reported by all health facilities in the two districts, at district level and disaggregated to the block-level; and the Expected Level of Achievement (ELA), which is projected data. They calculated the ELA data using the latest 2011 census and applied an average annual population growth rate using parameter values reported in the Annual Health Survey 2012–2013 (eg, crude birth rate) and in the HMIS data from previous years (eg, percentage of pregnancies leading to a stillbirth in 2015–16). We averaged these two data sources, the HMIS and the ELA, to obtain the administrative estimate.

A household probability survey was conducted during 7 September 2016–25 September 2016 in each district in a standardised manner[28 29] using Lot Quality Assurance Survey (LQAS). The LQAS survey is a stratified random sample used to assess health service coverage. The strata are administrative blocks (11 in Aurangabad, 14 in Gopalganj), which are subdistrict units in India. The strata were weighted by their population size. The first stage sample uses probability proportional to size to randomly sample villages (typically, n=19 per stratum).[28] In each random location a second stage sample identifies an index household using segmentation sampling[30 31] from locally constructed hand-drawn maps. The next closest household is then selected for interview to reduce the likelihood that households not included in the map do not have zero probability of selection. In the selected household, individuals in three target groups (women with children 0–2, 3–5 and 0–5 months of age[32] are listed with one selected randomly using a random number table. The remaining target group is selected in the next closest house using the same protocol.[33] The resulting sample of individuals is random and provides its own SE estimator

used for calculating a 95% CI. One member of each target group only was selected in a sampled village.

The total number of random samples for each indicator is 209 (Aurangabad) and 266 (Gopalganj). LQAS gives reliable estimates for indicators at the district level[34] and allows the classification of blocks, as 'performing adequately' or not. However, in this paper, we do not use the data for classification, but rather for estimation.

### Indicators for annealing

We based the selection of 10 indicators for annealing on the ease of combining HMIS data temporally to the survey data (table 1). For the LQAS survey, we measured ANC, institutional delivery and neonatal care outcomes in the population of women with children 0–2, 3–5 and 0–5 months of age[32] resulting in a total of 627 (Aurangabad) and 798 (Gopalganj) interviews. We used the HMIS data from the last 5 months of reports (April–August 2016) preceding the LQAS survey (tables 1 and 2), so that they matched temporally.

### The AT design

As in earlier work,[17] we apply AT in each sub-district for each indicator. The district-level estimators are an aggregation of appropriate block-level coverage data, weighted by the population sizes. Data analysis was done using Stata SE V.15 and R V.3.4.1.

#### Survey coverage estimator

For all but two indicators, the numerator of the LQAS coverage is the sum of correct responses in each of two target group (mothers of children 0–2 months and 0–5 months) in each block. The denominator is the sum of the denominators of both groups ($19 \times 2 = 38$). For the two indicators we used additional data from a third target group, mothers of children 3–5 months, where the denominator is the sum of the denominators of each target group ($19 \times 3 = 57$) (table 2).

A step in the annealing process requires a weighted average of component parts; the LQAS estimator is one such component. For each component, the weight used is inversely proportional to its variance. If the LQAS coverage, p, is away from the extremes of zero or one, we can use the standard binomial formula, $p(1-p)/n$. When the LQAS coverage is zero (or one), we calculate the weight by using a surrogate 'SE' from a 95% CI using Louis' approach[35] and dividing its length ($1-0.05^{(1/n)}$) by ($2 \times 1.96$), and squaring the resultant.

To assess any block-level clustering effect between target groups sampled in parallel, we use the two-tail McNemar's test for correlated proportions for each indicator,[36] across all blocks sampled in each district.

#### HMIS coverage estimator

The numerator of the HMIS coverage is reported in table 1 (column 4). The denominator is the average between the total number of ANC registration per month calculated by the ELA projected data and those reported

**Table 1** List of 10 indicators selected for applying the annealing technique in Bihar state, India

| Generic name of indicator | LQAS indicator | Targeted groups (LQAS surveys) | HMIS indicator | Targeted months (HMIS reported data) |
|---|---|---|---|---|
| First trimester registration | Proportion of mothers of 0–2 months old whose pregnancy was registered in the first trimester in the last pregnancy | 0–2 months<br>0–5 months | No of pregnant women registered within first trimester | Apri–August 2016 |
| Three or more ANC | Proportion of mothers of 0–2 months old who had three or more ANCs during their entire period of last pregnancy | 0–2 months<br>0–5 months | No of pregnant women received 3 ANC check-ups during pregnancy | April–August 2016 |
| Received TT1 | Proportion of mothers of 0–2 months who received TT1 during their last pregnancy | 0–2 months<br>0–5 months | No of pregnant women given TT1 during current pregnancy | April–August 2016 |
| Received TT2 | Proportion of mothers of 0–2 months who received TT2 during their last pregnancy | 0–2 months<br>0–5 months | No of pregnant women given TT2 or booster during current pregnancy | April–August 2016 |
| Received 100 IFA tablets | Proportion of mothers of 0–2 months who received at least 100 IFA tablets during their last pregnancy | 0–2 months<br>0–5 months | Total no of pregnant women given 100 IFA tablets | April–August 2016 |
| Institutional deliveries (all) | Proportion of 0–3 months infants who had institutional delivery (survey conducted September 2016) | 0–2 months<br>3–5 months<br>0–5 months | Total no of deliveries conducted in institutional setting (private and public) including C sections | April–August 2016 |
| Public Institutional Deliveries | Proportion of 0–3 months infants who had institutional delivery at public facility (survey conducted September 2016) | 0–2 months<br>3–5 months<br>0–5 months | No of deliveries conducted in public institutions including C sections | April–August 2016 |
| Baby weighed during delivery | Proportion of 0–2 months infants who were weighed during delivery | 0–2 months<br>0–5 months | No of new-borns weighed at birth | April–August 2016 |
| Early Initiation of breast feeding | Proportion of 0–2 months infants who were breast fed within an hour of their birth | 0–2 months<br>0–5 months | No of new-borns breast fed within 1 hour of birth | April–August 2016 |
| Postpartum home visit within 24 hours (home deliveries only) | Home visit by any FLW within 24 hours of delivery | 0–2 months<br>0–5 months | No of new-borns visited within 24 hours of delivery for deliveries conducted at home. | April–August 2016 |

ANC, antenatal care; FLW, front-line worker; HMIS, Health Management Information Systems; IFA, iron folic acid; LQAS, Lot Quality Assurance Survey; TT2, two or more tetanus toxoid vaccinations.

in the HMIS data (averaged over April–August 2016) (table 2).

To calculate the weight, as we did for the LQAS estimator, to use for the HMIS estimator, we use the fact that both estimators are estimating the same quantity, so their difference should be estimating zero. To construct a variance for the HMIS coverage estimator, we use the average mean square error (MSE), which is the average of the squared difference between the HMIS and LQAS coverage estimates over all the blocks composing one district. The MSE is calculated for the whole district, and can be decomposed in two ways:

$$MSE = \frac{1}{K} \sum_{k=1}^{K} \left( p_{HMIS\_k} - p_{LQAS\_k} \right)^2 \text{(first decomposition)}$$

$$= \sigma_{HMIS}^2 + \sigma_{LQAS}^2 \text{(second decomposition)}$$

where:
- $K$ is the total number of blocks in the district;
- $k$ is the index for the block in the concerned district, and ranges from one to $K$;
- $\sigma_{HMIS}$ is the SE of the HMIS coverage $p_{HMIS}$;
- $\sigma_{LQAS}$ is the SE of the LQAS coverage $p_{LQAS}$.

The first decomposition measures how much the HMIS and LQAS coverage estimators differ across the

**Table 2** Data characteristics

| Characteristics | HMIS | Household survey |
|---|---|---|
| Source | Bihar State Health Society | LSTM* study |
| Sample size | Aurangabad: 5917 (ELA) 5350 (HMIS) Gopalganj: 5952 (ELA) 6373 (HMIS) | Aurangabad (11 blocks): 19×11= 209 per target group Gopalganj (14 blocks): 19×14 = 266 per target group |
| Time period | April–August 2016 | 7–25 September 2016 |

ELA, expected level of achievement; HMIS, Health Management Information Systems; LSTM, Liverpool School of Tropical Medicine.

blocks composing the district. The second decomposition shows the MSE is also the sum of the two variances of HMIS and LQAS estimators, assuming each source of data was collected independently from each other. This decomposition also assumes that both HMIS and LQAS are unbiased, that is, that the expectation of both estimates is the same). We calculate the MSE using the first decomposition and subtract the LQAS variance to obtain the HMIS variance (and thus $\sigma_{HMIS}$). This can then be used to calculate the weight, as we show in[17] and in online supplemental information 1 and 2 tables S1-S9.

### Assessing variation in local-level data quality

To advance the use of AT in multiple settings we developed a tool for subdistrict-level analysis. We assess the variation in the quality of the HMIS indicators in 250 block-level comparisons of the 10 indicators $\left(10 \times \left(11 \; Aurangabad + 14 \; Gopalganj\right)\right)$ and 20 district-level comparisons. We classify the value of the HMIS coverage relative to the CI around the combined estimate. If the HMIS coverage is within 95% CI of $p_{combined}$ the cell is coloured green. If the HMIS estimate is within a reasonable percentage difference of each boundary of the CI, for example 10% (we colour those cells light red if it is above the high boundary or light blue if it is below the lower boundary), we are assured its value is not far from the survey estimate. While when the HMIS coverage is beyond the reasonable percentage difference of each boundary, the discrepancy might be too large to recommend combining the two estimates and to not use the HMIS estimate (we colour these dark red or dark blue).

The survey data were collected originally for assessment of a maternal and child healthcare programme in Bihar. We obtained oral rather than written informed consent from all respondents, because of the high illiteracy rate. The survey data we later treated as secondary data for the current study which was supported by our donor and the Bureau of Statistics for the Ministry of Health for the State of Bihar. After giving permission to access the State HMIS, they facilitated the collection of the HMIS data.

### Patient and public involvement

This study does not involve patients. Also, the public were not involved in the design, conduct and reporting of the research. The public was engaged as interviewees. To ensure local engagement all data capture was carried out in close coordination with the State Ministry of Health of Bihar. We also shared the results with them and offered further dissemination of results, and engaged them for data use and action planning activities.

## RESULTS

### Data characteristics and at characteristics

Among the 10 indicators, the MSE between the two sources of data ranges from 0.013 to 0.193 in Aurangabad, and between 0.005 and 0.391 in Gopalganj (see table 3 for the indicators). Three blocks in Gopalganj have an HMIS coverage larger than 100% (two or more tetanus toxoid vaccinations (TT2): 104% in Barauli and 103% in Kateya; iron folic acid (IFA) tablets distribution: 147% in Pach Deuri). The HMIS coverage averaged over blocks ranges between 12% and 80% in Aurangabad, and between 10% and 84% in Gopalganj. The LQAS coverage averaged over the blocks ranges between 3% and 98% in Aurangabad, and between 5% and 97% in Gopalganj. The McNemar tests did not detect block-level clustering except for the four tests with indicators measuring institutional delivery in all facilities or in public facilities.

The weighting factor used in the calculation of the combined estimator varies for each indicator in each district (figure 1). For six indicators (registration during first trimester of pregnancy, receiving TT1, receiving >100 IFA tablets, institutional delivery in any facility, deliveries in public facilities, baby weighed within 1 hour of birth), all block-level weighting factors $w$, are below 0.10 in both districts, indicating the contribution of HMIS estimator to the combined estimator is no higher than 10% (table 3 and online supplemental tables S1–S9). We use a one-tail test to check for difference of the weights between the two districts, based on the observed mean difference. The weights for Gopalganj are higher than Aurangabad's for three indicators: first trimester registration, receiving TT1 and having a postpartum visit—for all three, one-sided t-test p<0.01. The Gopalganj weighting factors are lower than Aurangabad's for the other seven indicators: >3 ANC visits, receiving TT2, receiving>100 IFA tablets, institutional delivery in any facility, deliveries in public facilities, early initiation of breastfeeding (EIBF), baby weighed at delivery (all seven one-sided t-test p<0.05). This result indicates that there are more discrepancies between the HMIS estimates and the LQAS estimates in Aurangabad when compared with Gopalganj for the first three indicators, while the discrepancies are greater in Gopalganj for the latter seven indicators. All but one of the weights for the indicator >3 ANC visits in Aurangabad are between 0.4 and 0.5, indicating an almost equal contribution of the HMIS and LQAS estimators to the resulting combined estimator. Alternatively, there are

**Table 3** Summary information before application of AT

| District | Indicator | MSE | Average value of over all blocks* | Average value of over all blocks* | McNemar test p value† |
|---|---|---|---|---|---|
| Aurangabad | Pregnancy registered during first 3 months | 0.105 | 0.66 | 0.42 | 0.47 |
| | three or more ANC | 0.013 | 0.64 | 0.62 | 0.46 |
| | Mothers who were protected against tetanus during pregnancy (Neonatal TT) | 0.152 | 0.61 | 0.98 | 0.71 |
| | Mothers who were protected against tetanus TT2 | 0.027 | 0.79 | 0.67 | 0.68 |
| | Mothers who received iron folic acid for 100 days or more when they were pregnant | 0.150 | 0.38 | 0.03 | 0.71 |
| | Institutional delivery (Both private and public facility) | 0.193 | 0.34 | 0.74 | 0–2 vs 3–5 months: 0.73 0–2 vs 0–5 months: 0.39 0–5 vs 3-5 months: 0.20 |
| | Institutional delivery (public facility) | 0.060 | 0.34 | 0.53 | 0–2 vs 3–5 months: 0.05 0–2 vs 0–5 months: 0.84 0–5 vs 3–5 months: 0.04§ |
| | Baby Weighed within 1 hour of birth | 0.078 | 0.50 | 0.67 | 0.65 |
| | Early Initiation of Breast feeding | 0.032 | 0.50 | 0.57 | 0.49 |
| | Home visit by any FLW within 24 hours of delivery | 0.018 | 0.12 | 0.06 | 0.18 |
| Gopalganj | Pregnancy registered during first 3 months | 0.078 | 0.72 | 0.48 | 0.86 |
| | Three or more ANC | 0.051 | 0.73 | 0.63 | 0.37 |
| | Mothers who were protected against tetanus during pregnancy (Neonatal TT) | 0.065 | 0.76 | 0.97 | 0.81 |
| | Mothers who were protected against tetanus TT2‡ | 0.031 | 0.84 | 0.78 | 0.91 |
| | Mothers who received iron folic acid for 100 days or more when they were pregnant‡ | 0.391 | 0.63 | 0.05 | 0.20 |
| | Institutional delivery (Both private and public facility) | 0.245 | 0.35 | 0.85 | 0–2 vs 3–5 months: 0.04§ 0–2 vs 0–5 months: 0.06 0–5 vs 3–5 months: 0.89 |
| | Institutional delivery (public facility) | 0.068 | 0.35 | 0.59 | 0–2 vs 3–5 months: 0.01§ 0–2 vs 0–5 months: 0.03§ 0–5 vs 3–5 months: 0.51 |
| | Baby weighed within 1 hour of birth | 0.117 | 0.44 | 0.75 | 0.05 |
| | Early Initiation of breast feeding | 0.051 | 0.45 | 0.65 | 0.17 |
| | Home visit by any FLW within 24 hours of delivery | 0.005 | 0.10 | 0.05 | 0.45 |

*11 blocks in Aurangabad and 14 blocks in Gopalganj.
†McNemar test comparing 0–2 vs 0–5 months unless otherwise specified.
‡Three blocks in Gopalganj have an HMIS coverage larger than 100% (TT2 indicator: 104% in Barauli and 103% in Kateya; IFA tablets indicator: 147% in Pach Deuri).
§The McNemar test reports a major difference at p<0.05.
ANC, antenatal care; AT, annealing technique; FLW, front-line worker; HMIS, Health Management Information Systems; IFA, iron folic acid; MSE, mean square error; TT2, two or more tetanus toxoid vaccinations.

large variations in the weights for the indicator measuring the one visit by any front-line worker (FLW) within 24 hours of delivery in Gopalganj; while, the MSE is small in this instance.

Comparing indicator estimates of 20 $p_{HMIS}$ vs $p_{LQAS}$ across the two districts, five differed by <0.10 and six others by <0.20. The remaining nine indicator estimates show larger differences (table 3; Online supplemental tables S1–S9).

### ANC services

All estimates, SEs and CIs are calculated at the block and district levels (as an illustration, see the detailed results for first trimester ANC registration in table 4; Online supplemental tables S1–S4 for other indicators). For the five indicators measuring coverage related to ANC and

birth preparedness, the combined block-level estimate differs from the LQAS estimate by 1% (received>100 IFA tablets) to 10% (≥3 ANC visits); the block-level combined estimate differs from the HMIS estimate at most by 28% (TT1) to 133% (received >100 IFA tablets). SEs for the HMIS data are 1.4 to 32.3 times larger than those calculated for the combined estimates.

### Institutional delivery

For the two indicators measuring the proportion of infants institutionally delivered (online supplemental tables S5 and S6), the combined block-level estimate differs from the LQAS estimate at most by 1% (all facilities) and 4% (public facilities only); the combined estimate differs from the HMIS estimate at most by 84% (all facilities) and 58% (public facilities only). SEs for the HMIS data

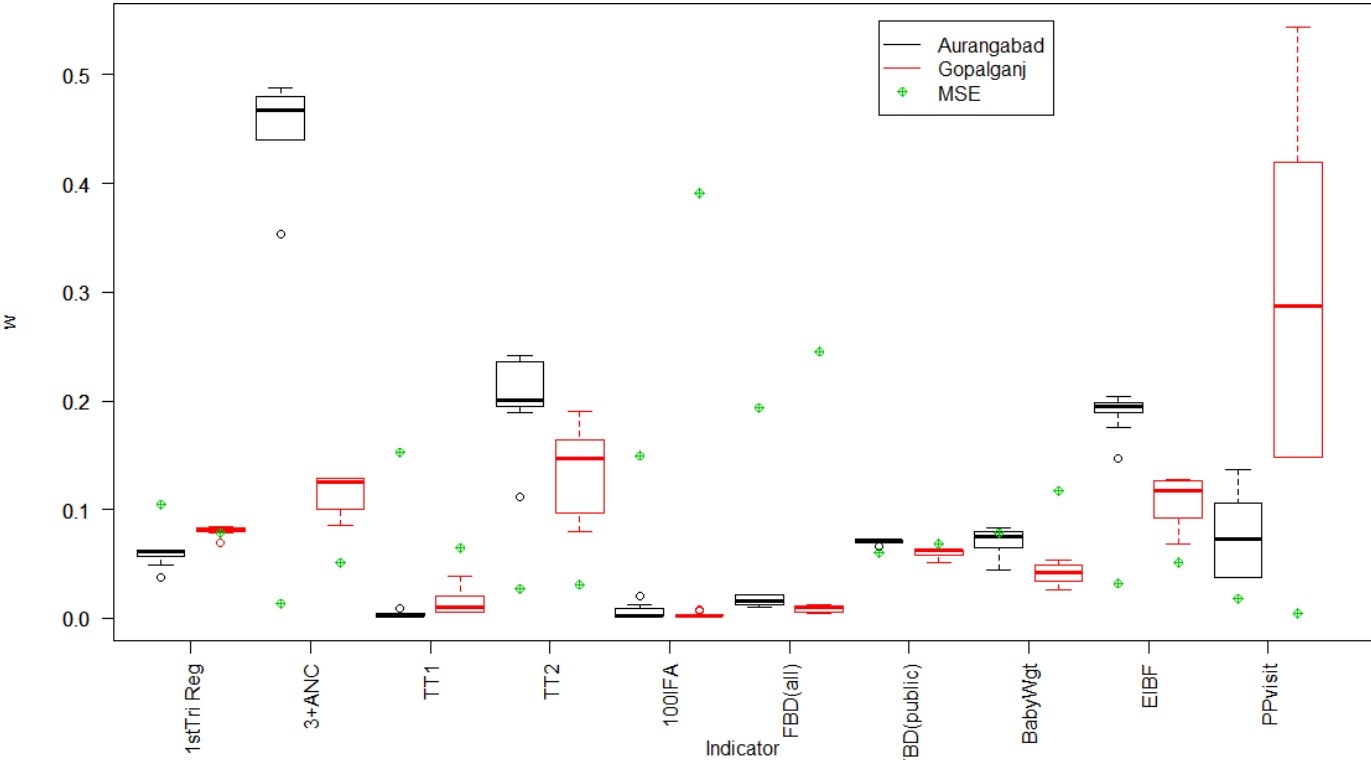

**Figure 1** Distribution and average of $w$ for each indicator and district. Circles with no filling indicate outlier values. ANC, antenatal care; EIBF, early initiation of breastfeeding; MSE, mean square error; TT2, two or more tetanus toxoid vaccinations; FBD, facility-based delivery.

are 3.7–14.6 times larger than those calculated for the combined estimates.

### Neonatal health

For the three indicators measuring coverages related to neonatal health (online supplemental tables S7–S9), the combined block-level estimate differs from the LQAS estimate at most by 3% (newborns who were visited by any FLW within 24 hours of home delivery) to 8% (EIBF); the combined estimate differs from the HMIS estimate at most by 36% (newborns who were visited by any FLW within 24 hours of home delivery) to 70% (EIBF). SEs for the HMIS data are 1.4 to 6.2 times larger than those calculated for the combined estimates.

### Local level variation

We first look at the two rows of figure 2 that summarise the district level measures and see that, on average, Aurangabad's HMIS measures (row 13) behave slightly better than Gopalganj's (row 28). We see that for two indicators both districts are red; for three they are both blue; for two they are both light red; and for two Aurangabad's are light blue vs Gopalganj's dark blue; and, for one Aurangabad's is green and Gopalganj's is light red.

We get a more detailed contrast when investigating the block-level results. Of the 250 comparisons 36.4% of the HMIS estimates are within the CI of the combined estimate with Aurangabad displaying greater accuracy (44.5%) than Gopalganj (30%). For two indicators (100 IFA tablets received and facility based delivery (all)) the

results are consistently poor across blocks. The same can be said of TT1 in Aurangabad, but not in Gopalganj. These indicators warrant further investigation into why the two sources of data have such discrepancies. The 3+ANC displays good agreement in Aurangabad, but a mixed evaluation in Gopalganj. Similarly, with the remaining indicators, we can detect subtly different results within the two districts. These results suggest substantial variation across blocks, districts and indicators in the HMIS accuracy.

### DISCUSSION

The results of our study show numerous discrepancies of block-level HMIS coverage estimates in both districts compared with the probability samples. However, the extent of these discrepancies varies across the blocks. We would expect consistency among block-level results given their geographical and political proximity within these two districts. Using the combined estimates as our guides, they differ from the respective probability estimates by at most 10%. In contrast, the combined estimates differ from the HMIS estimates by up to 84.2% (facility-based delivery (all) in Aurangabad's block Sadar), except for one instance where the difference is 133% due to the administrative coverage being >100% (distribution IFA during pregnancy in Pach Deuri). SEs for the HMIS estimates are between 1.4 and 32.3 times larger than for the

**Table 4** Women who registered for ANC during the first 3 months of pregnancy

| Block | $p_{HMIS}$ | $\sigma_{HMIS}$ | $p_{lqas}$ | $\sigma_{lqas}$ | $w$ | $p_{combined}$ | $\sigma_{combined}$ | 95% CI | $\sigma_{HMIS}/\sigma_{combined}$ |
|---|---|---|---|---|---|---|---|---|---|
| Aurangabad sadar | 0.42 | 0.31 | 0.39 | 0.08 | 0.06 | 0.4 | 0.077 | (0.25 to 0.55) | 4.08 |
| Barun | 0.55 | 0.31 | 0.58 | 0.08 | 0.06 | 0.58 | 0.078 | (0.43 to 0.73) | 4.04 |
| Daudnagar | 0.67 | 0.31 | 0.53 | 0.08 | 0.06 | 0.54 | 0.078 | (0.38 to 0.69) | 3.99 |
| Deo | 0.65 | 0.31 | 0.55 | 0.08 | 0.06 | 0.56 | 0.078 | (0.41 to 0.71) | 4.01 |
| Goh | 0.75 | 0.32 | 0.18 | 0.06 | 0.04 | 0.21 | 0.062 | (0.08 to 0.33) | 5.14 |
| Haspura | 0.46 | 0.31 | 0.55 | 0.08 | 0.06 | 0.55 | 0.078 | (0.39 to 0.70) | 4.01 |
| Kutumba | 0.78 | 0.31 | 0.42 | 0.08 | 0.06 | 0.44 | 0.078 | (0.29 to 0.60) | 4.04 |
| Madanpur | 0.74 | 0.31 | 0.32 | 0.08 | 0.05 | 0.34 | 0.073 | (0.20 to 0.48) | 4.29 |
| Nabinagar | 0.81 | 0.31 | 0.37 | 0.08 | 0.06 | 0.39 | 0.076 | (0.25 to 0.54) | 4.13 |
| Obra | 0.68 | 0.31 | 0.5 | 0.08 | 0.06 | 0.51 | 0.079 | (0.36 to 0.67) | 3.99 |
| Rafiganj | 0.77 | 0.32 | 0.26 | 0.07 | 0.05 | 0.29 | 0.070 | (0.15 to 0.42) | 4.53 |
| **Aurangabad district Average*** | **0.68** | **0.1** | **0.41** | **0.02** | | **0.42** | **0.023** | **(0.38 to 0.47)** | **4.23** |
| Baikunthpur | 0.73 | 0.27 | 0.53 | 0.08 | 0.08 | 0.54 | 0.078 | (0.39 to 0.70) | 3.46 |
| Barauli | 0.74 | 0.27 | 0.39 | 0.08 | 0.08 | 0.42 | 0.076 | (0.27 to 0.57) | 3.53 |
| Bhorey | 0.66 | 0.27 | 0.53 | 0.08 | 0.08 | 0.54 | 0.078 | (0.39 to 0.69) | 3.46 |
| Bijaipur | 0.81 | 0.27 | 0.47 | 0.08 | 0.08 | 0.5 | 0.078 | (0.35 to 0.65) | 3.46 |
| Gopalganj Sadar | 0.67 | 0.27 | 0.37 | 0.08 | 0.08 | 0.39 | 0.075 | (0.24 to 0.54) | 3.58 |
| Hathua | 0.94 | 0.27 | 0.71 | 0.07 | 0.07 | 0.73 | 0.071 | (0.59 to 0.87) | 3.81 |
| Kateya | 0.89 | 0.27 | 0.39 | 0.08 | 0.08 | 0.43 | 0.076 | (0.29 to 0.58) | 3.53 |
| Kuchaikote | 0.74 | 0.27 | 0.39 | 0.08 | 0.08 | 0.42 | 0.076 | (0.27 to 0.57) | 3.53 |
| Manjha | 0.67 | 0.27 | 0.61 | 0.08 | 0.08 | 0.61 | 0.076 | (0.46 to 0.76) | 3.53 |
| Pach Deuri | 0.49 | 0.27 | 0.47 | 0.08 | 0.08 | 0.47 | 0.078 | (0.32 to 0.63) | 3.46 |
| Phulwaria | 0.61 | 0.27 | 0.5 | 0.08 | 0.08 | 0.51 | 0.078 | (0.36 to 0.66) | 3.45 |
| Sidhwalia | 0.79 | 0.27 | 0.42 | 0.08 | 0.08 | 0.45 | 0.077 | (0.30 to 0.60) | 3.5 |
| Thawe | 0.67 | 0.27 | 0.42 | 0.08 | 0.08 | 0.44 | 0.077 | (0.29 to 0.59) | 3.5 |
| Uchkagaon | 0.71 | 0.27 | 0.42 | 0.08 | 0.08 | 0.44 | 0.077 | (0.29 to 0.59) | 3.5 |
| **Gopalganj district average*** | **0.73** | **0.08** | **0.48** | **0.02** | | **0.5** | **0.022** | **(0.46 to 0.54)** | **3.53** |

*The bold text and values are district averages across all the Blocks in Aurangabad and Gopalganj districts
ANC, antenatal care.

combined estimates (online supplemental tables S1, S9 and S4).

The HMIS data provides two sources of information to calculate the denominator (reported HMIS and ELA). Across the 14 blocks of Gopalganj, the two values were very close. In Aurangabad, we observe a linear relationship between the two sets of values, with the ELA values being slightly higher than the HMIS recorded values, suggesting the projected data overestimates the numbers reported. Since there was no evidence that one was better than the other, we chose to take the average of both values to define the denominator of the HMIS estimator. This step can be used in future applications.

A general attractiveness of the HMIS results for management purposes is their results apply at the block-level, and we do not wish to lose this property. We are fortunate because this level also corresponds to the strata in the LQAS surveys, and thus, this is the information level provided by the combined estimates. For greater granularity, LQAS is a favourable method to use for AT. In this study, we were also able to increase the sample size by pooling the data from two or three target populations surveyed in parallel within a block.

Our approach allows us to contrast the two districts of Bihar by studying the block-level (subdistrict-level) weighting factors $w$. For example, six indicators across both districts, have weights lower than 0.10, highlighting a large SE in the HMIS estimates vs those in the probability sample estimates which leads to a wide 95% CI in the HMIS estimates. The contribution of the HMIS estimator to the combined estimator for the indicator measuring visits by any FLW within 24 hours of birth estimator is higher in Gopalganj; the weights range between 0.15 and 0.54, indicating a more balanced contribution between

| Block or district | 1stTri Reg | 3+ANC | TT1 | TT2 | 100IFA | FBD(all) | FBD(public) | BabyWgt | EIBF | PPvisit |
|---|---|---|---|---|---|---|---|---|---|---|
| Aurangabad Sadar | green | green | blue | green | lightred | blue | blue | blue | blue | green |
| Barun | green | lightblue | blue | green | red | blue | blue | blue | blue | green |
| Daudnagar | green | green | blue | lightred | red | blue | green | green | green | green |
| Deo | green | green | blue | green | red | blue | green | green | green | green |
| Goh | red | green | blue | lightred | red | lightblue | green | green | green | green |
| Haspura | green | green | blue | lightred | green | blue | green | green | green | green |
| Kutumba | red | green | blue | green | red | blue | lightblue | lightblue | green | green |
| Madanpur | red | green | blue | lightred | red | blue | lightblue | green | lightred | lightred |
| Nabinagar | red | green | lightblue | green | red | blue | lightblue | green | green | red |
| Obra | lightred | green | blue | green | red | blue | blue | blue | green | lightred |
| Rafiganj | red | green | blue | green | red | lightblue | green | lightred | green | lightred |
| **Aurangabad District Average** | **red** | **green** | **blue** | **lightred** | **red** | **blue** | **blue** | **lightblue** | **lightblue** | **lightred** |
| Baikunthpur | lightred | lightred | blue | green | red | blue | lightblue | lightblue | green | green |
| Barauli | red | red | green | lightred | red | blue | lightblue | green | lightblue | green |
| Bhorey | green | green | lightblue | green | red | blue | green | green | lightblue | lightred |
| Bijaipur | red | green | lightblue | green | red | blue | blue | blue | blue | lightred |
| Gopalganj Sadar | red | lightred | green | lightred | red | blue | blue | blue | blue | green |
| Hathua | lightred | red | blue | lightred | red | blue | green | blue | green | green |
| Kateya | red | green | green | red | red | blue | lightblue | lightblue | green | green |
| Kuchaikote | red | red | lightblue | green | red | blue | blue | blue | green | green |
| Manjha | green | lightblue | lightblue | green | red | blue | blue | blue | lightblue | green |
| Pach Deuri | green | green | blue | blue | red | blue | lightblue | green | blue | green |
| Phulwaria | green | green | blue | lightblue | red | blue | green | lightblue | blue | green |
| Sidhwalia | red | green | blue | green | red | blue | blue | blue | lightblue | green |
| Thawe | lightred | lightblue | blue | lightred | red | blue | lightblue | blue | blue | green |
| Uchkagaon | red | green | lightblue | lightred | red | blue | blue | blue | green | lightred |
| **Gopalganj District Average** | **red** | **lightred** | **blue** | **lightred** | **red** | **blue** | **blue** | **blue** | **blue** | **lightred** |

**Figure 2** HMIS coverage value compared with the 95% CI of the $p_{combined}$ estimate for 10 Indicators. Five categories: (blue) HMIS coverage is more than 10% below the lower bound. (Light blue) HMIS coverage is less than 10% below the CI lower bound. (Green) HMIS coverage is within the CI. (Light red) HMIS coverage is less than 10% above the CI upper bound, (red) HMIS coverage is more than 10% above the CI upper bound. ANC, antenatal care; EIBF, early initiation of breastfeeding; HMIS, Health Management Information Systems; TT2, two or more tetanus toxoid vaccinations; FBD, facility-based delivery.

the convenience and the probability sample, meaning that some indicators have similar estimates while others do not. In Aurangabad the only indicator having a contribution weight of the HMIS estimator of over 0.35 in all blocks is three ANC visits. Hence, we are not observing a systematic discrepancy across indicators and districts. Some HMIS estimates are similar to the probability survey estimates for specific indicators, other indicators display similarity in only certain subdistricts or districts, while other indicators show consistent discrepancies between HMIS and survey result estimates. Ex post, we can learn from these concordance and discordance patterns how well some indicators are defined and measured across districts.

These Bihar-HMIS AT results differ from the Benin and Madagascar-CHD AT results. In the latter study the administrative data consistently over-estimated prevalence.[17] Such is not the case here as 25.6% of indicators overestimate and 38% underestimate coverage. Future research should map the variation in data quality by country and indicators as well as subnationally, and understand the reasons for the discrepancies between the HMIS and probability sample. These discrepancies might be due to one or more reasons: poor-quality denominators in the HMIS in the state or in specific districts; inaccurate or incomplete recording of the HMIS numerator data in one district or at the block level; or incomplete transmission of HMIS records from a health centre to the block level for aggregation.[37] It is also possible that

the survey and HMIS indicators are not measuring quite the same health system product. The perception of a respondent to a survey question in their own home and that of the health worker in a health centre may differ. Such indicators may need replacement as well. Related to this issue is a question of a more general nature. In situations when two estimates differ substantially, is it advisable to combine conflicting estimates? We think this issue can be answered by the data: as the estimates are different, a compromise suggested by the data is to give appropriate weight to the disparate estimates when combining them—a principle which we demonstrate. Our AT method is grounded on the average squared difference between the HMIS and LQAS estimates of the same quantity (see formula in online supplemental file 1). Two assumptions are essential for applying the formula: (1) the two estimates have the same expectation, that is, they measure the same indicator, (2) the two estimates are uncorrelated, which is guaranteed by their independent source of data collection. The first assumption requires that the target population, time period and geographical areas are the same in each source of information. The resulting weights assigned to each estimate illustrate the reliance on the quality of the other source of data in terms of variability. The more variable the survey estimate is, the higher the weight for the HMIS contribution to the combined estimate, and vice and versa. As to the prescription for future strengthening of the HMIS and DHIS2, this is a separate issue the solution

of which depends on information obtained from principled future research.

Our AT method uses statistical principles to combine data coming from different sources and calculates error for all sources as well as for the resulting combined estimate. This innovative product is valuable. The AT approach is different to WHO's well-intentioned attempt to improve HMIS estimates with Computation Logic; its estimates rely on professional judgements and do not produce error terms or CIs–a limitation noted by the authors.[38] Therefore, the risk associated with their resultant is unknown. The Institute for Health Metrics and Evaluation applies more complex quantitative analyses rather than just human judgements. It produces an estimate of the number of additional children covered or overestimated.[39] However, like computational logic, it does not report a measure of its error.

We are living in an information conscious era with continual demand for, and the ability to produce, more and better data[1 40] obtained from multiple sources. Using accurate information can improve a population's health and reduce waste. We guide the integration of existing data sources to provide a more complete assessment of national health and to reduce the cost to the health system. While we consider numerous data sources, HMIS, which generally do not produce SEs, are touted as having increasing importance for improving the coverage and quality of health services.[41 42] Yet, HMIS are not accurate. Improving and measuring the quality of health information systems is greatly needed.[43] Our examples exemplify a ubiquitous situation, namely, the difficulty with having a good estimator of the denominator. These problems occur in developed and low resource countries,[15] leading some policy-makers to call for a restructuring of information systems.[43] The current strategy for improving information involves rolling-out a computerised DHIS2. Though promising, whether the DHIS2 improves data quality is yet to be determined. Nevertheless, its concern is production of quality numerators. Our study shows that AT detected variation in the quality of the HMIS by indicator and sub-district location, and can be used to strengthen systems like the DHIS2 by identifying indicators and their location where improvement is needed. While it does not tell us the reasons for the errors it shows where they exist and their magnitude. This information can be used to understand and correct the sources of errors, which can improve health programmes and reduce waste.

Although we have discussed in detail the statistical principles of our approach, we should make clear its limitations that the global health community can address. Availability of survey data across multiple districts and subdistricts (Blocks in the case of India) is an issue. Survey data are not always widely available or conducted at a frequency that is needed for timely monitoring of health programmes. Concomitant survey data are essential for using AT to improve routine data. However, one single survey is not necessarily what is needed. Multiple surveys carried out at approximately the same time can be used. What is needed is a central data warehouse where surveys sponsored by government, international and civil society are accessible for use in AT, as are clear descriptions of sampling procedures and data codebooks. We are early in this era of improving data quality. While we have focused in this paper on developing appropriate statistical solutions, other health systems strengthening innovations are also needed. The international community needs to consider survey data as a public good. While their primary use may be for programme strengthening in a small catchment area, when considered together with other databases, survey data can have an enormous impact on improving data quality in the health system as a whole.

These results indicate the need to establish a service working side by side of the DHIS2 to inculcate hybrid estimation using AT as a standard component of district information systems. It should be an independent partner of the DHIS2, under the stewardship of lead international agencies such as UNICEF, the Global Fund to Fight AIDS, Tuberculosis and Malaria, USAID, World Bank and bi-lateral organisations (USAID, DFID). It would work independently and transparently to advance hybrid estimation not only in support of DHIS2, but also, and possibly more importantly, to support local public health practitioners and national policy-makers focus their programmes to address priorities.

## CONCLUSION

Our statistical innovation, hybrid prevalence estimation using statistical AT, requires concomitant data from a random survey sample[44] conducted at the subdistrict-level and subdistrict-level HMIS data. The LQAS survey in Bihar was undertaken for another purpose and so its use in AT was at no additional expense. Other statistical surveys, including the Demographic and Health Survey, are potential candidates. Provided both datasets are available in multiple sub-districts, we can estimate the variability of each indicator between subdistricts, and thus, construct not only a combined estimate but also its 95% CI at both subdistrict and district levels and a 95% CI for the HMIS estimator. With the new visual tool presented here, AT results can be quickly interpreted.

AT is intended to improve the quality and use of the HMIS while reducing waste due to using inaccurate information. This study shows that bringing data from existing household surveys together with HMIS data permits the calculation of more accurate and precise decentralised prevalence measures by combining HMIS and probability samples at very little cost. In addition to measuring coverage, it also allows us to evaluate the HMIS in practice and point to corrective measures. Such results lead to better systems for tracking the public's health.

**Acknowledgements** We gratefully acknowledge the essential roles of Hemant Das, Alok Prahdan and Sanjay Biswa for their careful field work supporting the

LQAS survey and the HMIS data retrieval. We thank Prof Imelda Bates, Prof Brian Faragher and Nancy Vollmer for their valuable feedback on an earlier version of this manuscript.

**Contributors** JJV and MP developed the research question; MP and CJ led the development of the mathematical statistics for the annealing technique; JJV and BD developed the survey design; JJV developed the survey methodology; BD managed the survey and data quality; CJ carried out the statistical analyses; JJV obtained the funding and donor support for the research; JJV, MP and CJ interpreted the data; CJ was responsible for data curation; all authors wrote and reviewed the paper; JJV acted as guarantor.

**Funding** This research was funded by the Bill & Melinda Gates Foundation Investment ID OPP1142889.

**Competing interests** None declared.

**Patient consent for publication** Not applicable.

**Ethics approval** The survey protocol was approved in 2016 by the Indian Institute of Public Health Bhubaneswar Institutional Ethics Committee (IIPHB-IEC-2016/010) and the Ethical Review Board of the Liverpool School of Tropical Medicine (research protocol 16-023RS).

**Provenance and peer review** Not commissioned; externally peer reviewed.

**Data availability statement** Data are available on reasonable request. Household survey data are available on reasonable request. The recurrent data belong to the Ministry of Health of the State of Bihar and requests for data must be made to them.

**ORCID iD**
Joseph J Valadez http://orcid.org/0000-0002-6575-6592

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
