## [Reviewer comments · BMJ Open]

ARTICLE DETAILS

TITLE (PROVISIONAL)	An innovative approach to improve information accuracy in a two-district cross-sectional study in Bihar, India
AUTHORS	Jeffery, Caroline; Pagano, Marcello; Devkota, Baburam; Valadez, Joseph

VERSION 1 – REVIEW

REVIEWER	Annibale Cois Stellenbosch University Faculty of Medicine and Health Sciences, Division of Health systems and Public Health
REVIEW RETURNED	29-May-2021

GENERAL COMMENTS	In this article, the authors present an application of a previously developed statistical technique (Annealing techniques, AT) to combine routine and survey estimates of 10 indicators assessing maternal and neonatal health services in two districts in India in 2016. Their stated objectives of 1) quantify the error in the routine estimates and 2) improve their accuracy. The article is built around a well-defined research question and, the overall study design is appropriate, the outcomes clearly defined and the results address directly the research question. Some aspects of the manuscript, however, would benefit from further clarification and integration, especially in the light of replicability in other contexts. My comments are the following: 1) Abstract • The setting and participants sections repeat information on study objectives and add other details that should probably go in the previous section, but provide very little understanding of the actual setting. 2) Strengths and limitations • Is it not clear the origin of the statement that survey data (that appears to be collected through multistage sampling with multiple levels of clustering) is not affected by design effect. Please, explain (in the methods).• Please, make more explicit what the “rigorous assessment” of the process entailed • Albeit the methods has been previously applied, its validity is still based on a series of assumptions that need to be acknowledged as limitations and discussed (see below). 3) Methods • A summary table with the distribution of the size and characteristics of the sample in the survey and the source of routine indicators would help to better understand the sources data, albeit the information is mostly reported discursively in the relevant paragraphs.• The description of the annealing techniques included in the
---

	manuscript and the additional material should detail and justify the basic assumptions underlying its applicability. In particular, the authors state that the second decomposition of the MSE in the formula on page 30 of the manuscript relies on the assumption of independence of the estimates (which they justify), but, (unless I am missing something), there is no mention to the fact that that decomposition of the MSE as the sum of two variances also requires the absence of bias (i.e. that the expectation of the two estimates are the same). Also, the assumption that the MSE calculated over the district also applies to each block deserves some comment. This formula is central to the procedure and its validity (or approximate validity) is the fundament that justifies the properties of the combined estimates, both in terms of bias and efficiency. I suggest that this explicitly discussed, and the consequences of its violations on the conclusions drawn in the manuscript explored.  • Please, justify the use of a 1-tails test for testing differences in weights. 4) Discussion A deeper discussion of strengths and limitations of the approach should be included in the discussion. Beyond the needs of assessing the validity of the assumption underlying the statistical techniques and their potential effects on the resulting estimates, the problem of availability of survey data across multiple districts seems central. Survey data are not so widely available and surveys are conducted at a frequency that is usually well below what is needed for timeously monitoring of health programmes. In absence of concomitant survey data, the method is not directly applicable to improve routine data, albeit still useful as a tool for a period evaluation of the level of reliability of these estimates. 5) References References 17 and 23 refer to the same publication.
--	--

REVIEWER	William Weiss USAID Bureau for Global Health, Maternal and Child Health and Nutrition
REVIEW RETURNED	11-Aug-2021

GENERAL COMMENTS	See attached pdf file with comments regarding "No" responses to the checklist above. The reviewer provided a marked copy with additional comments. Please contact the publisher for full details.
---

VERSION 1 – AUTHOR RESPONSE

Reviewer 1: Dr. Annibale Cois, Stellenbosch University Faculty of Medicine and Health Sciences

Comment 1:

In this article, the authors present an application of a previously developed statistical technique (Annealing techniques, AT) to combine routine and survey estimates of 10 indicators assessing maternal and neonatal health services in two districts in India in 2016. Their stated objectives of 1) quantify the error in the routine estimates and 2) improve their accuracy.

The article is built around a well-defined research question and, the overall study design is appropriate, the outcomes clearly defined and the results address directly the research question.

Authors Response: Thank you

Comment 2:

Some aspects of the manuscript, however, would benefit from further clarification and integration, especially in the light of replicability in other contexts.

1) Abstract

The setting and participants sections repeat information on study objectives and add other details that should probably go in the previous section, but provide very little understanding of the actual setting.

Author Response: We have now made it clear the setting is Bihar, India which is India's 5th poorest state with half the population living below the poverty line. We dropped the statement about the importance of routinely reported data as we address this point in the main text.

Comment 3: 2) Strengths and limitations

Is it not clear the origin of the statement that survey data (that appears to be collected through multistage sampling with multiple levels of clustering) is not affected by design effect. Please, explain (in the methods).

Authors' Response: The data are collected as a stratified random sample, where the strata are represented by the Supervision Areas. We have removed 'with no apparent design effect', and in the methods removed the words 'two-stage'.

Comment 4: Please, make more explicit what the "rigorous assessment" of the process entailed

Authors' Response: We changed this phrase to "has been assessed using a statistically principled approach"

Comment 5: Albeit the methods has been previously applied, its validity is still based on a series of assumptions that need to be acknowledged as limitations and discussed (see below).

3) Methods

A summary table with the distribution of the size and characteristics of the sample in the survey and the source of routine indicators would help to better understand the sources data, albeit the information is mostly reported discursively in the relevant paragraphs.

Authors' Response: Thank you for this suggestion. We have added the information we think you are requesting at the bottom of page 6 beginning at line 24. The new text is: "The current study assesses the transferability of AT to a new setting, HMIS data from Bihar state located in northeast India. Bihar is the 5th poorest state of India where half the population lives below the poverty line. It is one of the most densely populated states (N = 110million), and has some of the weakest maternal

and child health and nutrition indicators in India ^{25 26}. Both districts are primarily rural with a mix of upper caste and lower caste Hindu residents, and minority Muslim communities.”

Comment 6: The description of the annealing techniques included in the manuscript and the additional material should detail and justify the basic assumptions underlying its applicability. In particular, the authors state that the second decomposition of the MSE in the formula on page 30 of the manuscript relies on the assumption of independence of the estimates (which they justify), but, (unless I am missing something), there is no mention to the fact that that decomposition of the MSE as the sum of two variances also requires the absence of bias (i.e. that the expectation of the two estimates are the same). Also, the assumption that the MSE calculated over the district also applies to each block deserves some comment.

This formula is central to the procedure and its validity (or approximate validity) is the fundament that justifies the properties of the combined estimates, both in terms of bias and efficiency. I suggest that this explicitly discussed, and the consequences of its violations on the conclusions drawn in the manuscript explored.

Authors Response:

In Supplementary Information 1, we added the sentence ‘This decomposition also assumes that both HMIS and LQAS are unbiased, i.e. that the expectation of both estimates is the same).’

Comment 7: Please, justify the use of a 1-tails test for testing differences in weights.

Authors Response: The test performed is two-tail. We have clarified this in the Methods section.

Comment 8: 4) Discussion

A deeper discussion of strengths and limitations of the approach should be included in the discussion.

Beyond the needs of assessing the validity of the assumption underlying the statistical techniques and their potential effects on the resulting estimates, the problem of availability of survey data across multiple districts seems central. Survey data are not so widely available and surveys are conducted at a frequency that is usually well below what is needed for timeously monitoring of health programmes. In absence of concomitant survey data, the method is not directly applicable to improve routine data, albeit still useful as a tool for a period evaluation of the level of reliability of these estimates.

Authors Response: Agree, the method we describe here refers to annealing two sources. If, as the reviewer states, we only have one source of data, we cannot anneal it with a non-existent other. We have developed methods for using concomitant survey data available for proximal areas—such as in the same district, but not in the same commune. Our paper on the topic is currently under review: Extrapolating Hybrid Estimation to Districts with Missing Probability Survey Data, A Ocampo, JJ Valadez, B Hedt-Gautier, and M. Pagano.

Comment 9: 5) References

References 17 and 23 refer to the same publication.

Authors Response: Thank you. This is corrected.

Reviewer 2 Dr. William Weiss, USAID Bureau for Global Health

Comment 1: Title Consider alternative title, this is not proven in the article, although intuitive.

Authors Response: Thank you for your interesting suggestion. May we please suggest that it is this intuitive assertion is what stimulates the work in this arena for us and for others such as some in the Health Data Collaborative. We have developed an alternative title: "An innovative approach to improve information accuracy in a two-district cross-sectional study in Bihar, India "

Comment 2: Page 2 line 14 for the reader, consider here stating this is a secondary data analysis of survey data and HMIS data, all datasets are without personal identifiers.

Authors Response: You are indeed correct. We have added this point to the methods section where we have a higher word limit. Please see page 6 line 18: "... and two sources of data collected for purposes other than this study. HMIS data from the two districts and a probability survey data collected as part of an earlier assessment of maternal and child health care coverage in Bihar²³. This secondary data analysis uses datasets without personal identifiers."

Comment 3: Pg 3, Line 8: Is this really shown in this paper... what determines readiness for international use vs. additional testing?

Authors Response: We take your point, but also think that a benefit of having a statistically principled method gives confidence to it, while donors provide funding for further testing. We fully support this. In the meantime we have modified this sentence to read "... an effective method ready for additional international assessment while also being used to provide affordable information to improve health services."

Comment 4: Pg 3, Line 11: Use same font throughout

Authors Response: Done

Comment 5: Pg 3, line 20-21: This seems correct but contradicts the statement (in likely interpretations) about the technique's readiness for international use.

Authors Response: See response to comment 3.

Comment 6: Pg 4 line 9: More accurate to say that they are done at 3-5 year intervals. therefore, surveys are not sufficient in frequency for many decisions. The term elaborate is subjective. Surveys are resource intensive but do not deplete all resources (as may be inferred by the statement).

---The insufficiency of HH surveys is probable sufficient to make the point here.

Authors Response: We have modified line 9 to read: "... household surveys are generally considered resource intensive, and being carried out at 3-5 year intervals are not sufficiently frequency for many decisions.

Comment 7: Line 11: Perhaps more accurate to say, "more frequently used for decision-making and for annual reviews than HH surveys." It may not be the basis for many decisions, however, as decisions are made based on census, human resource information systems, and logistics, pharmacy information systems.

Authors Response: We modified line following your good suggestion. The sentence now reads: "As a result, data generated routinely through the Health Management Information Systems (HMIS) is more frequently used for decision-making and for annual reviews than household surveys."

Comment 8: Line 14: HMIS data is biased toward those who access health services, but that does not mean it cannot be representative under the circumstances that health facilities are the only source of some services, and the denominator (e.g., from census) is fairly accurate.

Authors Response: We see the point you are making, however, as you know, in cases where services are provided more generally and a person has a choice of providers, it may not be clear which numerator should be used with which denominator. The situation is even more complex when private providers are used whose facility records are not always included in the DHIS2. Similarly, the delays in reporting HMIS to central levels are well known. Take Nigeria, Uganda, Mozambique as examples. Regarding the denominator: a recent census may be fairly accurate, however, in many LMIC experiencing in and out migration they soon lose their accuracy. In line 14 we were attempting to make a more simple point, namely, that HMIS are subject to many errors which users need to be aware of.

Comment 9: Line 15: and where there are competing services in the community or nearby homes that do not report into the HMIS

Authors Response: By addressing comment 8 we add more context which may also address your previous comment. Line 15 now reads "... social determinants, and in settings where there are competing services in the community that do not report into the HMIS. "

Comment 10: Line 19: Very much out of date... many resolutions around data since. Health Metrics Network no longer exists. There exists a newer, but different Health Data Collaborative.

Authors Response: Thank you. We have updated the text of line 23 which now reads, "... This was followed by other resolutions and in 2015 by a high-level summit on Measurement and Accountability for Results leading to the formation of the Health Data Collaborative (HDC) in March 2016 and supported by global partners. HDC's mandate is more extensive than that of HMN, having an ultimate objective of aiding countries to improve the quality and availability of health data and their ability to consistently and accurately report on progress towards the health-related Sustainable Development Goals."

Comment 11: Line 24: over 70 at this point... out of date

Authors Response: we updated to the figure on the DHIS2 website, namely, 73+.

Comment 12: Pg 5, line 14: What about efforts to improve denominators through the census?

Authors Response: Could the reviewer provide references for the efforts he refers to?

Comment 13: Line 21: Can't be narrower if one does not exist for HMIS?

Authors Response: We deleted the word "narrower" and modified the sentence to read, "... It builds on our earlier pioneering work in 2011¹⁶ and refined in 2018¹⁷ that provides a coverage estimator with a 95% Confidence Interval (CI) by combining data from a probability survey and HMIS data. HMIS coverage estimates have not before had 95%CI to inform users of their accuracy. ..."

Comment 14: Pg6 line 4: This statement would be better acknowledging recent effort to use surveys to improve HMIS (see 2 examples below)... what appears unique about AT is the ability to generate uncertainty intervals around HMIS.

Maina, I., et al. (2017). "Using health-facility data to assess subnational coverage of maternal and child health indicators, Kenya." Bull World Health Organ **95**(10): 683-694.

Simmons, E. M., et al. (2020). "Assessing coverage of essential maternal and child health interventions using health-facility data in Uganda." Popul Health Metr **18**(1): 26.

Authors Response: Thank you for these suggestions. We added a sentence so the section now reads. "However, concurrent well-designed surveys have been a surprisingly neglected sources of information for improving HMIS ^{21 22}. Recent efforts have used surveys to improve HMIS estimates^{23 24}, although the resulting revised estimates do not have a corresponding 95%CI."

Comment 15: Line 23: Probably more accurate to say the tally information from paper registers and send a paper report of those tallies to BPHCs.

Authors Response: We edited the sentence as you recommend. "Staff at the HSCs tally information from paper registers and send a paper report to BPHCs where it is entered into a computerized HMIS system, and subsequently maintained electronically."

Comment 16: Page 7 Lines 6-12: This is an uncommon scenario. What was the rationale for averaging the two data sources? On the limited information here it does not seem justified to use the ELA data, as this is not typical in many countries this technique would be potentially generalizable. Please explain... and please comment on what would be expected in other settings.

Authors' Response: On the general principle that we have two estimates of the same quantity and no belief that one is preferable to the other, then a sensible approach is to average them. If both are unbiased, then this results in a more precise estimate. The same principle that says, take more observations. We could easily have ignored one or the other and it would not have made any difference to the methodology.

Comment 17: Page 12 line 20: Was survey done for this analysis? Or was the survey, done for other purposes, re-used for this analysis. If so, then this is a secondary data analysis with assumed permissions from respondents for the first survey? Please re-check the manuscript and update the ethical procedures to differentiate what was done for this analysis vs. the primary data collection... what was primary data collection for this manuscript vs secondary data analysis needs more clarity.

Authors' Response: We elaborated the Ethics Statement to include the information you requested. "The survey data were collected originally for assessment of a maternal and child health care programme in Bihar. The survey protocol was approved in 2016 by the Indian Institute of Public Health Bhubaneswar Institutional Ethics Committee (IIPHB-IEC-2016/010) and the Ethical Review Board of the Liverpool School of Tropical Medicine (research protocol 16-023RS). We obtained oral rather than written informed consent from all respondents, because of the high illiteracy rate. The survey data we later treated as secondary data for the current study which was supported by our

donor and the Bureau of Statistics (BoS) for the Ministry of Health for the State of Bihar. After giving permission to access the State HMIS, the BoS facilitated the collection of the HMIS data.”

VERSION 2 – REVIEW

REVIEWER	William Weiss USAID Bureau for Global Health, Maternal and Child Health and Nutrition
REVIEW RETURNED	07-Nov-2021
GENERAL COMMENTS	The changes are very helpful.